# Identification of Plasma Growth Factors and Cytokines as Diagnostic Biomarkers for the Lafora Form of Progressive Myoclonus Epilepsy

**DOI:** 10.3390/ijms26115354

**Published:** 2025-06-03

**Authors:** Mireia Moreno-Estellés, María Machio, Laura González, Marta Albuixech, Laura Abraira, Manuel Quintana, Manuel Toledo, Marina P. Sánchez, José M. Serratosa, Pascual Sanz

**Affiliations:** 1Centro de Investigación Biomédica en Red de Enfermedades Raras (CIBERER)—Instituto de Salud Carlos III, 46010 Valencia, Spain; 2Instituto de Biomedicina de Valencia, CSIC, Jaime Roig 11, 46010 Valencia, Spain; malbuixech@ibv.csic.es; 3Neurology Department, Instituto de Investigación Sanitaria Fundación Jiménez Díaz, Madrid Autonomous University (IIS-FJD, UAM), 28040 Madrid, Spain; maria.machio@quironsalud.es (M.M.); laura.gonzalez@quironsalud.es (L.G.); msanchezg2@hotmail.com (M.P.S.); joseserratosa@me.com (J.M.S.); 4Epilepsy Unit, Neurology Department, Vall d’Hebron University Hospital, Vall d’Hebron Research Institute, Vall d’Hebron Barcelona Hospital Campus, 08035 Barcelona, Spain; laura-abraira@vallhebron.cat (L.A.); maquintavh@gmail.com (M.Q.); manuel.toledo@vallhebron.cat (M.T.)

**Keywords:** Lafora disease, diagnostic biomarker, plasma samples, high-throughput screening

## Abstract

Lafora progressive myoclonus epilepsy (LD, OMIM#254780, ORPHA:501) is an ultra-rare and severe autosomal recessive neurological disorder that typically manifests in early adolescence. It is characterized by the accumulation of insoluble forms of aberrant glycogen in the brain and peripheral tissues. Given the urgent need for reliable tools to monitor disease progression, we aimed to identify reliable biomarkers in minimally invasive fluids, which could also provide valuable insights into the natural history of the disease. Plasma-EDTA samples from eleven LD patients and healthy controls were analyzed to identify potential biomarkers of LD using a high-throughput assay. The findings were subsequently validated using specific enzyme-linked immunosorbent assays (ELISAs). Eleven cytokines and growth factors were identified to be significantly reduced in LD patient samples compared to healthy controls. Among these, four mediators [platelet-derived growth factor subunit B (PDGF-BB), epidermal growth factor (EGF), brain derived growth factor (BDNF), and macrophage migration inhibitory factor (MIF)] exhibited the greatest fold change between the groups and were further validated. Given the minimally invasive nature of plasma sampling and the straightforward quantification via ELISA assays, these biomarkers hold strong promise for rapid translation to the clinic, potentially enhancing early diagnosis and longitudinal disease monitoring in LD patients.

## 1. Introduction

Lafora disease (LD, OMIM#254780, ORPHA:501) is an ultra-rare and fatal autosomal recessive form of progressive myoclonus epilepsy. LD typically manifests in early adolescence and is characterized by the accumulation of insoluble deposits of aberrant glycogen (polyglucosans) in the brain and peripheral tissues [1]. More recently, neuroinflammation has been defined as a second hallmark of LD in mouse models [2,3]. LD results from mutations in the EPM2A gene, which encodes the glucan phosphatase laforin, or in the EPM2B gene, which encodes the E3-ubiquitin ligase malin. Laforin and malin form a functional complex where laforin recognizes substrates to be subsequently ubiquitinated by malin [4]. This shared molecular mechanism likely explains the similar clinical presentations observed in patients carrying mutations in either gene. The laforin–malin complex negatively regulates glycogen synthesis by modulating the activity of glycogen synthase [5], thereby providing a mechanistic basis for the enhanced glycogen synthesis observed in LD patients.

There is no treatment for Lafora disease yet, although several strategies are now being developed. An approach consisting of the use of antibody–enzyme fusion protein based on recombinant alpha-glucosidase (VAL1221), to digest Lafora bodies, has been applied to several Italian patients. However, the results did not work as expected [6]. Perhaps, the most advanced strategy right now is the use of antisense oligonucleotides (ASOs), to diminish the levels of glycogen synthase (ION283, Phase 1 clinical trial NCT06609889) (https://clinicaltrials.gov, accessed on 30 March 2025). Unfortunately, there are no reports yet on the beneficial effects of this innovative treatment.

Clinically, LD is characterized by generalized tonic-clonic seizures, myoclonus, absences, and visual hallucinations. The disease progresses rapidly with patients experiencing cognitive deterioration, dementia, and increased seizure frequency, often leading to death within 5 to 15 years after onset due to complications such as status epilepticus or aspiration pneumonia [7]. The rarity of the disease (prevalence of less than 4/1,000,000 individuals) has hindered comprehensive studies of its natural history. However, a recent analysis of 298 LD cases provided valuable insights into key clinical milestones, including the mean age of onset (approximately 13 years), median survival time (around 11 years), and median time to loss of autonomy (approximately 6 years), among other parameters [7]. These findings complement earlier prospective studies using electroencephalographic (EEG) biomarkers and the Clinical Disability Progressive Scale (CDPS), which delineate five clinical stages of LD: stage 0, asymptomatic; stage I, with visual seizures only; stage II, with mild cognitive decline; stage III, with an established dementia and status epilepticus; stage IV, with myoclonic encephalopathy; and stage V, with progressive neurological deterioration, eventually leading to death from respiratory failure (Delgado-Escueta et al., 2015; Cure Lafora Epilepsy Meeting, Istanbul, Turkey, https://chelseashope.org/wp-content/uploads/2018/08/Istanbul-Meeting-Report-Escueta.pdf, accessed on 30 March 2025) [8].

To monitor disease progression, there is an urgent need to identify surrogate biomarkers associated with LD. These biomarkers, whether diagnostic or prognostic, could predict disease manifestations, track progression, and assess the efficacy of emerging therapies. To date, to our knowledge, only imaging techniques such as positron emission tomography (PET), magnetic resonance spectroscopy and imaging (MRI), and proton magnetic resonance spectroscopy (MRS) have been used to monitor the progression of LD patients [9,10,11,12].

Using animal models of LD, alternative potential biomarkers have been explored. For example, metabolomics analyses have revealed differences in brain metabolites between LD and control mice [13,14]. Additionally, our group recently identified elevated levels of specific inflammatory cytokines in the serum samples of LD mice compared to controls [3].

The aim is this work is to identify in the plasma of LD patients some determinants that could serve as diagnostic biomarkers of LD.

## 2. Results

### 2.1. Analysis of CXCL10, S100B, and CCL20 Protein Levels

In a recent study, we reported elevated levels of CXCL10, S100B, and CCL20 proteins in the blood serum of a mouse model of LD (Epm2b-/- mice) aged 12 months or older [3]. To assess whether these proteins were also elevated in human samples from LD patients, we measured their levels using ELISA assays in plasma samples from seven LD patients (LD1 to LD7) and four age-matched healthy controls (H1 to H4) (Table 1 and Appendix A). In the case of CXCL10, we observed higher levels of this chemokine in the LD samples compared to controls (median 64 pg/mL in healthy vs. 70 pg/mL in LD patients), but this difference was not statistically significant (*p* = 0.494; Appendix A). No significant differences were found either in the levels of S100B (median 1 ng/mL for healthy vs. 1 ng/mL for LD patients; *p* = 0.999; Appendix A) or in the case of CCL20 proteins (median 31 pg/mL for healthy vs. 1 pg/mL for LD patients; *p* = 0.530; Appendix A). These results indicate that the levels of CXCL10, S100B, and CCL20 are not reliable biomarkers for LD in humans.

### 2.2. Analysis of Human Plasma Samples Using High-Throughput Arrays

To expand our search for potential biomarkers of LD, we analyze the cytokine profile of plasma samples from LD patients using the Proteome Profiler Human Cytokine Array kit (see Section 4). This analysis included eight healthy samples (marked with $ in Table 1; four men and four women, with a range of age from 23 to 34 years) and the first extraction (T1) from eleven LD samples (four men and seven women, with an interval of age from 7 to 32 years), as described in Table 1.

The intensity of the signals was analyzed using the Quick Spot software provided by the manufacturer. In Appendix A, we show the average intensity of two spots for each mediator, the fold change value between healthy and LD samples, and the corresponding *p*-values. In Figure 1A, we show the heatmap corresponding only to those proteins with a fold change value between healthy and LD samples greater than 2 or lower than 0.5. A clear distinction between healthy and LD samples was observed for these proteins. In the right panel of Figure 1A, we indicate the fold change of the average intensity of each protein and the corresponding *p*-values. In this approach, we also confirmed the absence of significant differences in the levels of CXCL10 and CCL20 proteins between healthy and LD samples (see values in grey in Appendix A; the human array did not contain S100B antibodies).

In the LD samples, we only observed a significant increase in the case of GDF-15, a protein belonging to the transforming growth factor beta superfamily, which plays a role in regulating inflammatory pathways and is upregulated in cardiovascular and neuroplastic disorders [15,16,17] (Figure 1A and Appendix A). However, it has been reported that treatment with metformin increases the levels of GDF-15 [18], and since the LD patients with higher levels of GDF-15 in the array (e.g., LD1, LD4, LD5, LD08, LD9, LD10, and LD11; Figure 1A) were undergoing metformin treatment (see Appendix A), we decided to defer GDF-15 for further analysis.

Interestingly, we observed significant decreases in the levels of growth factors and other mediators in the LD samples (Figure 1A and Appendix A). Among them, we found particularly striking the decrease in the levels of platelet-derived growth factor AB/BB (PDGF-AB/BB; 8.45-fold decrease), PDGF-AA (5.07-fold decrease), and epidermal growth factor (EGF; 5.01-fold decrease). In addition, we found a decrease in the levels of chemokine CXCL15/ENA78 (3.18-fold decrease); chemokine CCL17/TARC (2.82-fold decrease); brain-derived neurotrophic factor (BDNF; 2.79-fold decrease); macrophage migration inhibitory factor (MIF, 2.45-fold decrease); thrombospondin-1, a glycoprotein that mediates cell-to-cell and cell-to-matrix interactions (2.36-fold decrease); angiopoietin-1, a glycoprotein that mediates interactions between the endothelium and surrounding matrix (2.25-fold decrease); Dkk-1, an antagonist of the Wnt/b-catenin pathway (2.17-fold decrease); and endothelial plasminogen activator inhibitor 1 (serpin E-1), which is an inhibitor of tissue-type plasminogen activator (tPA) and urokinase (uPA), which is regulator of cell migration (2.0-fold decrease) (the description of the proteins has been made according to Uniprot: https://www.uniprot.org/ (accessed on 30 March 2025). Then, we used the STRING v12.0 software to analyze the possible functional interaction between these 11 mediators. This analysis pointed out a potential hub of interactions involving PDGF-AA, PDGF-BB, EGF, and thrombospondin-1 (Figure 1B). Notably, the levels of none of these markers were decreased in samples from Epm2b-/- mice [3], pointing out again differences between LD mice and LD patients.

Among all these proteins, we decided to focus our attention on those with the highest average fold change in healthy vs. LD samples and lowest *p*-values, selecting PDGF-AB/BB, EGF, BDNF, and MIF for further validation (Figure 1A and Appendix A).

### 2.3. Confirmation of the Levels of PDGF-AB/BB, EGF, BDNF, and MIF by ELISA Assays

Next, we validated the levels of the four selected proteins mentioned above using individual ELISA assays. We analyzed eleven samples from both the healthy donors and the LD patients. In the case of the LD patients, since in some of them we obtained several samples corresponding to different times of the progression of the disease (T1 to T3), we plotted in Figure 2 the values corresponding to the first time of extraction (T1) (see Appendix A for the complete dataset of values).

First, we confirmed significantly lower levels of PDGF-BB protein in all the samples of LD patients in comparison to healthy controls (Figure 2A) (median of 2424 pg/mL in healthy samples vs. 215 pg/mL in LD patients; *p*-value < 0.0001). Secondly, we verified the reduction in the levels of the EGF protein in all the samples of LD patients (Figure 2B) (median of 210 pg/mL in healthy samples vs. 3 pg/mL in LD patients; *p*-value < 0.0001). In the case of BDNF, the decrease in the levels of this protein was also statistically significant in LD samples (Figure 2C) (median of 141 ng/mL in healthy samples vs. 49 ng/mL in LD patients; *p*-value = 0.0128). Finally, in the case of the MIF protein, we also found a decrease in the levels of this protein in LD samples (Figure 2D), which was statistically significant (median of 375 ng/mL in healthy samples vs. 26 ng/mL in LD samples; *p*-value = 0.0005).

### 2.4. Correlation of PDGF-BB, EGF, BDNF, and MIF Levels with the Clinical Presentation of LD Patients

To determine whether these four proteins could be considered biomarkers of LD, we performed a receiver operating characteristic (ROC) analysis to assess their predictive accuracy. The ROC curves analysis shown in Figure 3 showed excellent diagnostic capabilities for PDGF-BB, EGF, and MIF, with elevated values of AUC in all the cases (95.9% for PDGF-BB, 100% for EGF, and 100% for MIF). The optimal cut-off values were <904 pg/mL for PDGF-BB, <81 pg/mL for EGF, and <160 ng/mL for MIF. BDNF exhibited a moderate predictive value (AUC was only 81% and the optimal cut-off value was <70 ng/mL).

Next, we studied the possible correlation between the levels of these biomarkers and the progression of the disease in each LD patient. To this end, we plotted the absolute values of all the extractions of the LD samples (Figure 4). In the case of PDGF-BB, although the levels found in healthy samples showed substantial variability, the levels found in LD samples showed reduced values in most of the cases. In the case of the EGF and MIF proteins, the levels found in all LD samples were always reduced in comparison with those of the healthy subjects. However, in the case of BDNF, some values of LD samples were as high as some found in healthy controls. These results pointed out the better performance of PDGF-BB, EGF, and MIF as possible biomarkers of LD.

These analyses also demonstrated that in none of the cases, PDGF-BB, EGF, BDNF, and MIF, did we find any correlation between the levels of these mediators and the progression of the disease in each patient (samples at latter stages of disease did not show consistent changes with respect to values obtained in previous stages). In addition, values obtained at the first visit to the Neurology Department were mostly similar in all the patients, independently of the age of the patient, with some exceptions in the case of BDNF levels (Appendix A).

The results shown in Figure 4 also indicated that in the case of PDGF-BB, EGF, and MIF, there was no correlation between the levels of these mediators and the severity of the clinical presentation of the disease (Appendix A). Patients with a severe presentation (e.g., LD-02 and LD-03 patients, in stage IV, according to the Delgado-Escueta scale [8]) showed lower levels of these mediators than controls, and the same was true in the case of patients with asymptomatic or with very weak presentation (e.g., LD-06 and LD-07 patients, in stage 0 of the Delgado-Escueta scale [8]). Moreover, no differences in clinical presentations and values of the analyzed mediators were found between patients carrying mutations in the EPM2A or EPM2B genes, or between males and females. Importantly, we found lower levels of PDGF-BB, EGF, and MIF in samples from LD patients that were asymptomatic or presented mild symptoms (e.g., LD-06, LD-07, in stage 0; Appendix A), suggesting the potential utility of these biomarkers for early diagnosis.

### 2.5. Specificity of PDGF-BB, EGF, BDNF, and MIF as Biomarkers for LD

To determine whether the decreased levels of PDGF-BB, EGF, BDNF, and MIF were specific to LD, we analyzed samples from patients suffering from other monogenic epilepsies, including Dravet syndrome disease (caused by a pathogenic genetic variant in the sodium channel SCN1A gene) and GLUT1 deficiency syndrome (caused by a pathogenic genetic variant in the glucose transporter SLC2A1 gene) (Table 1). As shown in Figure 5, the three Dravet samples (OME-01 to OME-03) exhibited reduced levels of PDGF-BB (median of 1067 pg/mL vs. 2424 pg/mL in healthy samples; Figure 5A), EGF (median of 32 pg/mL vs. 210 pg/mL in healthy samples; Figure 5B), BDNF (median of 29 ng/mL vs. 141 ng/mL in healthy samples; Figure 5C), and MIF (47 ng/mL vs. 375 ng/mL in healthy samples; Figure 5D) (Appendix A). In the case of the two GLUT1 samples (OME-04 and OME-05), we found similar levels of PDGF in healthy samples (median 2887 pg/mL vs. 2424 pg/mL in healthy samples), but lower levels of EGF (median 130 pg/mL vs. 210 pg/mL in healthy samples), BDNF (median of 48 ng/mL vs. 141 ng/mL in healthy samples), and MIF (median 121 ng/mL vs. 375 ng/mL in healthy samples) in comparison with healthy samples (Figure 5A–D) (Appendix A). Unfortunately, the reduced number of samples precluded us from regular statistical analysis of these models.

## 3. Discussion

One of the main challenges in the field of rare diseases is their delayed diagnosis. The identification of specific biomarkers has become an urgent need since they could facilitate both the diagnosis and prognosis of such conditions. The main goal of this study was to identify potential blood biomarkers of LD. Using LD animal models, our group previously identified elevated levels of several chemokines and cytokines in the serum of Epm2b-/- mice [3]. However, when we analyzed the levels of these mediators (CXCL10, CCL20, and S100B) in plasma samples from a cohort of eleven LD patients, we found no statistical differences in comparison to the healthy controls. Differences in the phenotype presentation of the disease in mice and humans probably account for this discrepancy. To expand our search for potential LD biomarkers, we conducted a high-throughput analysis using an array of 111 spotted antibodies targeting various cytokines, chemokines, and growth factors. This study identified eleven mediators whose levels were statistically different from LD patients and healthy controls. Interestingly, the most prominent differences, accompanied by robust statistical significance, corresponded to growth factors, including PDGF-AA/BB, EGF, BDNF, and MIF. In all these cases, we found a decrease in the levels of these mediators in LD samples in comparison to healthy controls. Notably, the levels of none of these markers were decreased in samples from Epm2b-/- mice [3], pointing out again differences between mice and human.

Then, we selected four of these markers, PDGF-AB/BB, EGF, BDNF, and MIF, and validated their levels by ELISA. PDGF is a neurotrophic factor (NTF) produced by pericytes and endothelial cells [18]. Several isoforms of this protein exist, including PDGFA and PDGFB, which form homo and heterodimers (PDGF-AA, PDGF-AB, and PDGF-BB) [19,20]. These ligands act on PDGF receptors (PDGFRs) expressed in endothelial cells, neurons, astrocytes, and microglia [21]. As NTFs, they regulate neuronal function, and decreased PDGF-BB levels have been associated with neuronal loss and other disease-related outcomes [22]. For instance, a PDGF-BB knockdown impairs hippocampal neurogenesis and increases susceptibility to chronic stress in mice [23]. Consequently, PDGF-BB is considered a peripheral marker of neurodegeneration [24]. Lower plasma PDGF-BB levels correlate with mild cognitive impairment in Alzheimer’s disease (AD), PDGF-BB being considered one of the most important biomarkers associated with AD [20]. Similarly, reduced PDGF-BB levels in plasma have been observed in severe ischemic stroke patients [25]. In Pompe disease, a particular type of glycogen storage disorder due to impairments in the activity of the lysosomal alpha-glycosidase enzyme, lower levels of PDGF-BB in plasma are also detected, and they may be used to differentiate between asymptomatic and symptomatic patients [26]. Notably, this latter case is particularly relevant to LD, as both conditions involve glycogen accumulation, although in LD, polyglucosans accumulate in the cytosol rather than in the lysosome. Therefore, reduced levels of PDGF-BB in LD patients could indicate impaired neuronal functionality, neurodegeneration, and heightened hyperexcitability, as previously reported in animal models of LD [3]. Furthermore, PDGF-BB plays critical roles in angiogenesis, vessel stabilization, blood flow regulation, tissue repair, and blood–brain barrier (BBB) integrity. Deficient PDGF-BB levels disrupt these processes [27], potentially contributing to the increased T-lymphocyte infiltration into the brain parenchyma that occurs in LD mice [3]. Since exogenous administration of PDGF-BB enhances vascular health and reduces spontaneous epileptiform activity [27], we hypothesize that PDGF-BB treatment could benefit LD patients.

EGF is another NTF involved in cell survival, proliferation, migration, and differentiation. It is synthesized in both the brain and peripheral tissues [28], with its peripheral production gaining access to the central nervous system (CNS) through the blood–brain barrier, so the effect of EGF on the CNS is the sum of the CNS and peripheral production, the latter gaining access to CNS through the blood–brain barrier (BBB). EGF signals through the EGF receptor (EGFR), which is expressed on neurons, astrocytes, oligodendrocytes, and microglia, supporting neural stem cell maintenance, oligodendrocyte differentiation, and neuronal homeostasis [29]. In astrocytes, EGF regulates the glutamate–glutamine cycle by inducing glutamine synthase, which converts glutamate into glutamine for neuronal conversion to glutamate or GABA [29]. Reduced levels of EGF have been linked to demyelination, which could cause multiple sclerosis (MS) [30]. Decreased levels of EGF in plasma have also been reported in Parkinson’s (PD) and AD, limiting in this way EGFR activation [30]. Interestingly, EGF shares some signaling mediators with the PDGF pathway [31], and physical interactions between the EGFR and PDGFR signaling pathways have been proposed [31]. These results are in agreement with our STRING analysis, which suggests a functional link between PDGF and EGF (Figure 1B). Therefore, reduced levels of EGF in LD may contribute to neuronal dysfunction, neurodegeneration, and hyperexcitability.

BDNF is essential for neuronal plasticity, proliferation, differentiation, and connectivity. It functions also as a stress-responsive protection factor involved in learning and memory. BDNF signaling through the TrkB receptor promotes the expression of antioxidant enzymes, such as superoxide dismutase 2 (SOD2) and glutathione reductase, thereby protecting neurons and glial cells from excitotoxicity and oxidative damage [19]. Decreased levels of BDNF in plasma have been reported in Huntington’s disease (HD) and Gaucher disease (GD) [24]. The reduced levels of BDNF observed in LD patients may reflect their heightened susceptibility to oxidative stress, as reported in LD mouse models [32].

Finally, MIF is primarily produced by T-lymphocytes but is also synthesized by endothelial cells and neurons. It has a dual role in cellular pathophysiology. On the one hand, MIF is considered a pro-inflammatory cytokine that recruits multiple inflammatory mediators, leading to the activation of microglia and astrocyte-derived neuroinflammation [33]. But, on the other hand, MIF has a neuroprotective role in defending neurons from oxidative stress and apoptosis [33,34]: in PD, MIF has a neuroprotective effect by suppressing inflammatory responses, inhibiting apoptosis, and inducing autophagy [35,36,37]. Similarly, MIF protects against protein misfolding in stroke and amyotrophic lateral sclerosis (ALS) [37]. Thus, the reduced levels of MIF in LD could exacerbate neurodegeneration and neuronal hyperexcitability.

Our longitudinal study reveals a significant reduction in these key neuroprotective growth factors (PDGF-BB, EGF, BDNF, and MIF) in the plasma of LD patients compared to healthy controls. Since in the case of BDNF, some values of LD samples were as high as some found in healthy controls, in our opinion, the levels of PDGF-BB, EGF, and MIF show a better performance as possible biomarkers of LD. Interestingly, reduced levels of these biomarkers were observed in asymptomatic patients, highlighting their potential utility as early diagnostic indicators.

In this work, we have also analyzed three samples from Dravet and two from GLUT1 deficiency patients. Although sample sizes were insufficient for robust statistical analysis, these samples also showed reduced levels of some of the markers found in LD patients, suggesting common pathophysiological mechanisms. Given the crucial roles that these proteins have in neuronal survival, function, and homeostasis, their depletion may directly contribute to the neuronal dysfunction, neurodegeneration, and hyperexcitability characteristic of these types of genetic epilepsies (LD, Dravet, and GLUT1 deficiency).

We are aware that the main limitation of this study is the reduced number of independent samples. This is a general limitation for rare diseases, especially in the case of ultra-rare diseases such as Lafora disease, with a prevalence of less than four patients per 1,000,000 individuals, where access to LD samples is quite complicated.

## 4. Materials and Methods

### 4.1. Study Design

Plasma-EDTA samples from eleven LD patients and thirteen healthy subjects were provided by Fundación Jiménez Díaz, Hospital Vall d’Hebron, and the CIBERER Biobank. Healthy control samples were collected in two different recruitments (H samples). Samples from five other monogenic epilepsies (Dravet and GLUT1 deficiency, OME samples) were included to assess the specificity of the newly identified biomarkers (Table 1). Samples were processed following standard operating procedures and with the appropriate approvals from the corresponding Ethics and Scientific Committees (see Ethics approval statement). Healthy and OME subjects were selected to match the age range of the LD patients. When possible, LD samples were collected from the same patients during three consecutive visits to the Neurology Department, at different intervals (T1 to T3). Male and female LD patients with mutations in either the EPM2A or EPM2B genes were included in the study (Table 1). For each LD patient, a comprehensive battery of cognitive and behavioral assessments (including seizures, dementia, and gait impairment) was conducted (Appendix A), as well as the identification of the specific LD mutation. The Clinical Disability Progressive Scale (CDPS) (Delgado-Escueta et al., 2015; Cure Lafora Epilepsy Meeting, Istanbul, Turkey, https://chelseashope.org/wp-content/uploads/2018/08/Istanbul-Meeting-Report-Escueta.pdf, accessed on 30 March 2025) [8] was used to determine the stage of the disease at each visit (Appendix A). Data on ongoing treatments were also recorded.

Blood samples were collected in EDTA tubes, centrifuged at 800× *g* for 10 min at room temperature, and plasma aliquots were frozen at −80 °C until use.

### 4.2. Biomarker Validation

Selected mediators were validated using specific ELISA assays, including PDGF-BB (Abcam, Cambridge, UK, ab184860), EGF (Abcam, ab217772), BDNF (Cloud-Clone Corporation, Wuhan, China, SEA011Mi), MIF (R&D systems Minneapolis, MN, USA, DMF00B), CXCL10/IP10 (Abcam, ab173194), S100B (Abcam, ab234573), and CCL20/MIP3a (R&D systems, DM3A00). Manufacturer protocols were followed in each case.

### 4.3. High-Throughput Screening of Inflammatory Mediators

Plasma-EDTA samples from LD patients and healthy controls were analyzed using the Proteome Profiler Human Cytokine Array kit (R&D systems). This array consists of 111 different captured antibodies spotted on a nitrocellulose membrane for the detection of multiple cytokines, chemokines, growth factors, and other soluble proteins. The intensity of the signals was analyzed using the Quick Spot software 25.6.03 provided by the manufacturer (Appendix A). The analysis of the differential expression between healthy and LD samples was performed by using the FLASKI software (http://flaski.age.mpg.de, accessed on 30 March 2025). In this work, we focus our attention on mediators with a fold change between healthy and LD samples greater than 2 or less than 0.5 for further analysis.

### 4.4. Protein Interactions

Putative interactions among the selected mediators were assessed using STRING v12.0 software (https://string-db.org/, accessed on 30 March 2025), which integrates data from numerous sources, including experimental studies, computational predictions, and publicly available literature.

### 4.5. Statistical Analysis

Statistical differences between the groups were analyzed using unpaired, non-parametric Mann–Whitney *t*-tests, using GraphPad Prism version 6.0 statistical software (La Jolla, CA, USA). Results are expressed as medians with ranges. Statistical significance was defined as *p*-values **** *p* < 0.0001, *** *p* < 0.001, and * *p* < 0.05. Receiver operating characteristic (ROC) analysis was performed for the validation of the four putative biomarkers to assess their capability to diagnose LD by plotting sensitivity versus specificity, using the IBM SPSS statistical package (version 25; SPSS Inc., Armonk, NY, USA). The area under the ROC curve (AUC) was calculated and Youden Index was used to determine optimal cut-off points for each biomarker.

## Figures and Tables

**Figure 1 ijms-26-05354-f001:**
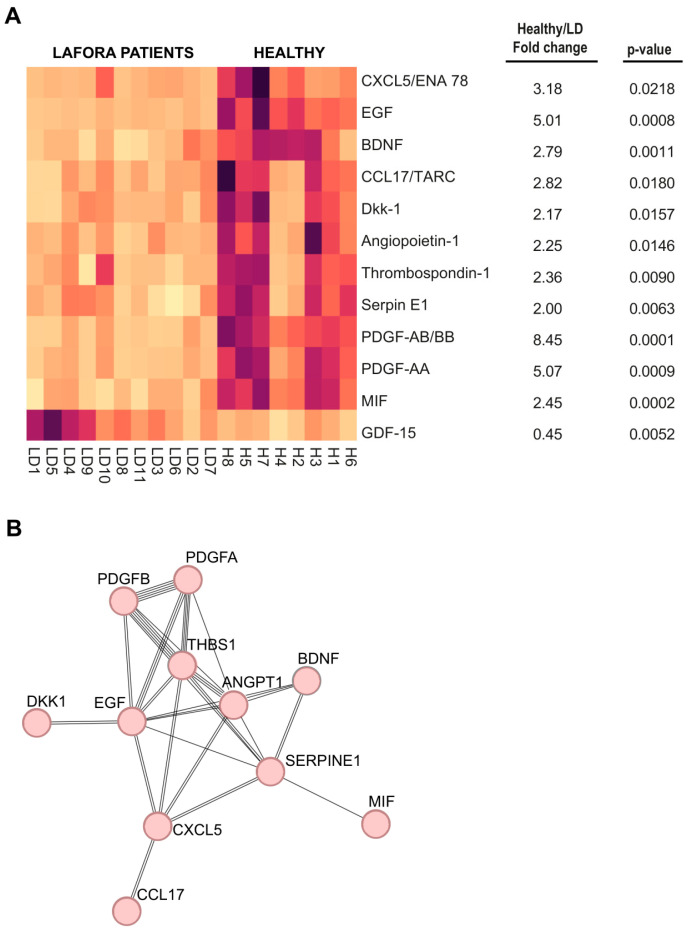
Differentially expressed proteins in Lafora disease patients and protein–protein interaction analysis. (**A**) Heatmap showing the levels of different proteins detected using the Proteome Profiler Human Cytokine Array kit (R&D systems, see Section 4). Plasma samples from eight healthy subjects and eleven LD patients were analyzed according to the manufacturer’s instructions. The intensity of the signals was analyzed using the Quick Spot software 25.6.03 provided by the manufacturer. Heatmap includes only proteins with a fold change value between healthy and LD samples greater than 2 or lower than 0.5, along with their corresponding *p*-values. (**B**) STRING analysis of the proteins identified in Figure 1A. Lines indicate possible protein–protein interactions, with a higher number of lines representing stronger predicted interactions.

**Figure 2 ijms-26-05354-f002:**
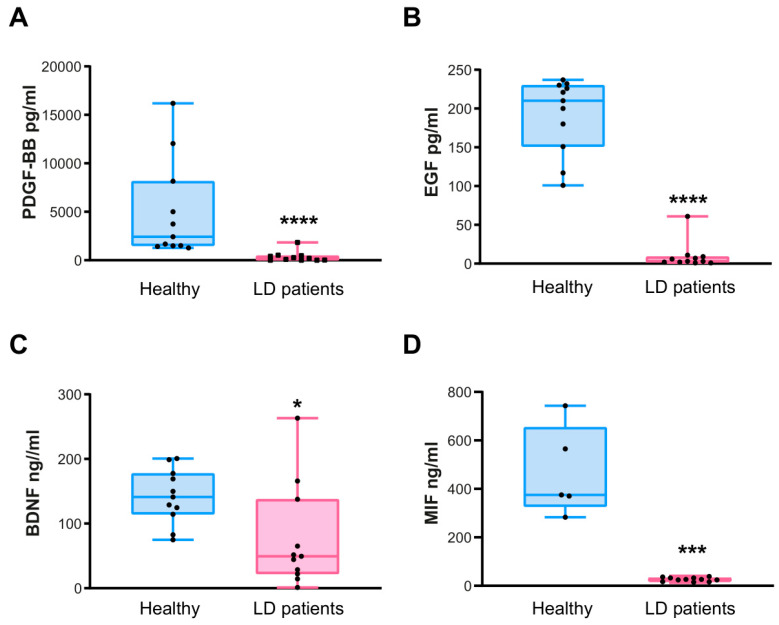
ELISA validation of the most significant proteins differentially expressed in LD samples. Plasma levels of PDGF-BB (**A**), EGF (**B**), BNDF (**C**), and MIF (**D**) in samples from eleven healthy subjects (pale blue) and eleven LD patients (pale pink) were analyzed by ELISA assays (for the MIF analysis, only five healthy subjects were included). Results are expressed as a median with a range. Statistical differences between the groups were assessed using the Mann–Whitney non-parametric *t*-test. *p*-values have been considered as **** *p* < 0.0001, *** *p* < 0.001, and * *p* < 0.05.

**Figure 3 ijms-26-05354-f003:**
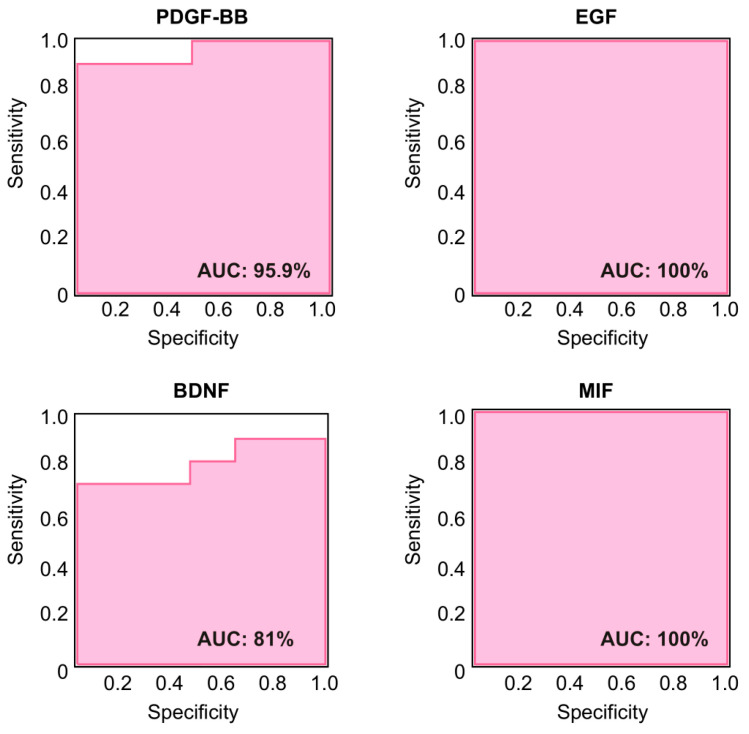
Receiver operating characterization (ROC) analysis of PDGF-BB, EGF, BDNF, and MIF. The ROC curves corresponding to the analyzed proteins were generated to predict their potential use as biomarkers. The area under the curve (AUC) is indicated in each case.

**Figure 4 ijms-26-05354-f004:**
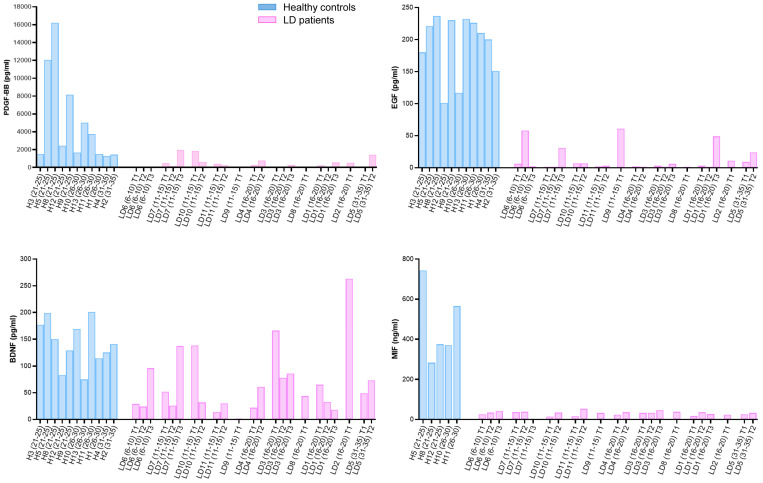
Longitudinal analysis of the levels of the PDGF-BB, EGF, BDNF, and MIF over time. Plasma samples from patients collected at different times of exploration were evaluated using ELISA analysis and compared to the values found in samples from healthy controls. The number of the LD sample, the age range at the time of the extraction, and the number of the extraction are indicated in each case. Some patients provided three samples, while others contributed with only two or one. Results are expressed as absolute values and plotted according to age.

**Figure 5 ijms-26-05354-f005:**
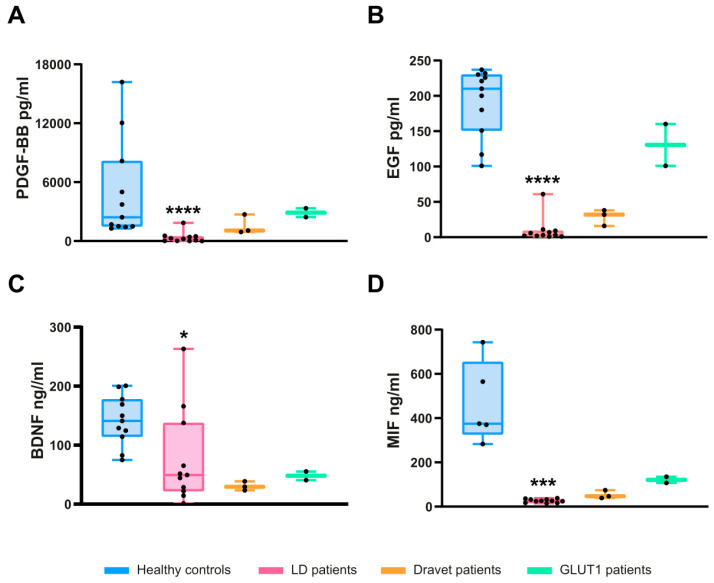
Comparison with other monogenic epilepsies. Plasma levels of PDGF-BB (**A**), EGF (**B**), BDNF (**C**), and MIF (**D**) were analyzed in three Dravet syndrome (pale orange) and two GLUT1 deficiency syndrome patients (pale green). For comparison, values from healthy controls and LD samples (as shown in Figure 1) were included in the plot. Due to the limited sample size, statistical analysis to assess differences between genotypes could not be performed. *p*-values have been considered as **** *p* < 0.0001, *** *p* < 0.001, and * *p* < 0.05.

**Table 1 ijms-26-05354-t001:** Age range and gender of the subjects analyzed in this work. F: female; M: male. T: time of the visit. $, healthy samples used in the Proteome Profiler Human Cytokine Array kit; LD, Lafora disease; OME, other monogenic epilepsy.

Healthy Subjects	Age Range (Years)/ Gender	Patient (Gene Mutated)	Age Range (Years) at the Beginning of the Study (T1)/Gender	Samples Obtained During the Study
H1 ^$^	26–30/F	LD1 (*EPM2A*)	16–20/F	T1, T2, T3
H2 ^$^	31–35/M	LD2 (*EPM2A*)	16–20/M	T1
H3 ^$^	21–25/F	LD3 (*EPM2B*)	16–20/F	T1, T2, T3
H4 ^$^	31–35/M	LD4 (*EPM2A*)	16–20/M	T1, T2
H5 ^$^	21–25/F	LD5 (*EPM2A*)	31–35/M	T1, T2
H6 ^$^	26–30/M	LD6 (*EPM2A*)	6–10/F	T1, T2, T3
H7 ^$^	21–25/F	LD7 (*EPM2A*)	11–15/M	T1, T2, T3
H8 ^$^	21–25/M	LD8 (*EPM2B*)	16–20/F	T1
H9	21–25/F	LD9 (*EPM2A*)	11–15/M	T1
H10	26–30/F	LD10 (*EPM2B*)	11–15/F	T1, T2
H11	26–30/M	LD11 (*EPM2B*)	11–15/F	T1, T2
H12	21–25/F			
H13	26–30/M			
		OME01 (SCNA1)	16–20/M	T1
		OME02 (SCNA1)	16–20/M	T1
		OME03 (SCNA1)	26–30/M	T1
		OME04 (SLC2A1)	26–30/M	T1
		OME05 (SLC2A1)	26–30/M	T1

## Data Availability

Data will be available upon request.

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
