# Peer review of "Identification of Plasma Growth Factors and Cytokines as Diagnostic Biomarkers for the Lafora Form of Progressive Myoclonus Epilepsy"

_ijms, 2025, doi:10.3390/ijms26115354_

Round 1

Reviewer 1 Report

Comments and Suggestions for Authors

The manuscript titled “Identification of Plasma Growth Factors and Cytokines as Diagnostic Biomarkers for the Lafora Form of Progressive Myoclonus Epilepsy” addresses the identification of diagnostic biomarkers for Lafora disease (LD), a rare autosomal recessive neurodegenerative disorder. The authors identified and validated four potential biomarkers—PDGF-AB/BB, EGF, BDNF, and MIF—using a cytokine array platform and ELISA. Notably, PDGF-BB, EGF, and MIF demonstrated strong diagnostic performance in ROC analyses, highlighting their potential utility in early disease detection. While the study is generally well-conceived and the findings are promising, several critical issues should be addressed to strengthen the manuscript:

  1. The small sample size significantly limits the statistical power and generalizability of the findings. A larger cohort would enhance the reliability and robustness of the results and provide stronger support for the proposed biomarkers.
  2. Although altered plasma levels of specific biomarkers in LD patients are reported, the underlying molecular mechanisms remain unexplored. Investigating these biomarkers further using cell-based or animal models would help elucidate their role in disease pathogenesis.
  3. The study would benefit from collecting samples at multiple time points, particularly across different stages of disease progression. This could provide insights into the dynamic changes of biomarkers over time and their correlation with disease severity.
  4. Consistent formats should be used in all the figures. The Fig4 and Fig5 shown different formats.
Comments on the Quality of English Language

The English could be improved to more clearly express the research.

Author Response

Comments and Suggestions for Authors. The manuscript titled “Identification of Plasma Growth Factors and Cytokines as Diagnostic Biomarkers for the Lafora Form of Progressive Myoclonus Epilepsy” addresses the identification of diagnostic biomarkers for Lafora disease (LD), a rare autosomal recessive neurodegenerative disorder. The authors identified and validated four potential biomarkers—PDGF-AB/BB, EGF, BDNF, and MIF—using a cytokine array platform and ELISA. Notably, PDGF-BB, EGF, and MIF demonstrated strong diagnostic performance in ROC analyses, highlighting their potential utility in early disease detection.

            We want to thank the reviewer for his/her positive comment on our manuscript.

While the study is generally well-conceived and the findings are promising, several critical issues should be addressed to strengthen the manuscript:The small sample size significantly limits the statistical power and generalizability of the findings. A larger cohort would enhance the reliability and robustness of the results and provide stronger support for the proposed biomarkers.

            We thank the reviewer for this comment. As indicated in the manuscript, Lafora disease is an ultra-rare disease with less than 4 affected per 1,000,000 individuals. At the Reference Center for Lafora disease in Spain (Fundación Jiménez Díaz), we did our best to obtain samples from 11 patients. Although, in numbers, the sample size seems small, we were able to identify some biomarkers that were differentially expressed in these LD samples.

Although altered plasma levels of specific biomarkers in LD patients are reported, the underlying molecular mechanisms remain unexplored. Investigating these biomarkers further using cell-based or animal models would help elucidate their role in disease pathogenesis.

            We thank again the reviewer very much for this comment. That is our next goal. So far we have checked the presence of these biomarkers in the blood from an animal model of LD (Epm2b-/-), and they were not differentially expressed. At present, we are on our way to obtaining human cellular models of LD to explore whether we can observe changes in these biomarkers.

The study would benefit from collecting samples at multiple time points, particularly across different stages of disease progression. This could provide insights into the dynamic changes of biomarkers over time and their correlation with disease severity.

            As indicated in Table S1, in some of the patients we collected samples at three different states of the disease. For example, patient LD1 progressed from stage III to stage IV; patient LD3 progressed from stage IV to stage V; patient LD11 progressed from stage I to stage II. As we indicated in the manuscript, we did not observe changes in the levels of the identified biomarkers related to the progression of the disease.

Consistent formats should be used in all the figures. The Fig4 and Fig5 show different formats.

            We have changed the format of Fig. 4 to the one of the rest of the Figures.

Comments on the Quality of English Language. The English could be improved to more clearly express the research.

            We have used ChatGPT to enhance language clarity. This has been included in the Acknowledgments section.

Reviewer 2 Report

Comments and Suggestions for Authors

Authors address a very rare and severe form of epilepsy that, like many other rare diseases, are not vastly investigated in the scientific community. They propose a somehow novel approach to diagnose and monitor Lafora disease using a non-invasive methodology, something that, mostly in patients’ perspective, are highly beneficial. However, there are some issues that authors need to address before publication:  

  • The abstract seems too long. Maybe authors should consider reducing conclusions length on it
  • In introduction, authors refer to EEG biomarkers to delineate stages of LD but, later, EEG is not included as a technique used to monitor disease progression. Please explain.
  • Which emerging therapeutics are currently available/under clinical trials? Some examples could be included in the introduction
  • Why choose patients with Dravet syndrome and GLUT1 deficiency to study specificity of the biomarkers? Shouldn't samples from patients with more common types of epilepsy have been included?
  • Please explain why total blood and serum were also not analyzed and, consequently, other types of biomarkers rather than those present in plasma. In fact, in the first sentence of the results section, authors refer “In a recent study, we reported elevated levels of CXCL10, S100B, and CCL20 proteins in the blood serum of a mouse model of LD”
  • By analyzing Figure S1, it seems to me that a lot of variances occurred between measurements. Can that be explained? Could that be important for further interpretation rather than just analyzing the median?
  • The authors state that PDGF-BB levels in healthy samples showed substantial variability. The same seems to happen in BDNF samples of LD patients. How can that be explained?
  • In the discussion, care must be taken with abbreviations early mentioned.
  • Considering that the upmost targets biomarkers that were found altered on LD patients in this study (PDGF-BB, EGF, BDNF, and MIF) are also changed in other neurological diseases, sometimes hardly diagnosed in early stages – Alzheimer’s disease, ALS, Parkinson’s disease, multiple sclerosis - how can these biomarkers be robust for LD?

Author Response

Comments and Suggestions for Authors. Authors address a very rare and severe form of epilepsy that, like many other rare diseases, are not vastly investigated in the scientific community. They propose a somehow novel approach to diagnose and monitor Lafora disease using a non-invasive methodology, something that, mostly in patients’ perspective, are highly beneficial.

            We want to thank the reviewer for his/her positive comment on our manuscript.

However, there are some issues that authors need to address before publication: 

The abstract seems too long. Maybe authors should consider reducing conclusions length on it.

            We have shortened the abstract as suggested.

In introduction, authors refer to EEG biomarkers to delineate stages of LD but, later, EEG is not included as a technique used to monitor disease progression. Please explain.

            The EEG technique was an additional test included in the Clinical Disability Progressive Scale (CDPS) described by Dr. Delgado-Escueta, which we have used to identify the severity of the disease. This was described at the Introduction.

Which emerging therapeutics are currently available/under clinical trials? Some examples could be included in the introduction

            We thank this reviewer's comment very much. We have included a paragraph at the Introduction indicating the currently emerging therapeutics. Now it says: “There is no treatment for Lafora disease yet, although several strategies are now being developed. An approach consisting of the use of antibody-enzyme fusion protein based on recombinant alpha-glucosidase (VAL1221), to digest Lafora bodies, has been applied to several Italian patients. However, the results did not work as expected (Muccioli et al., 2024; BMJ Open 14: e085062). Perhaps, the most advanced strategy right now is the use of antisense oligonucleotides (ASOs), to diminish the levels of glycogen synthase (Phase 1 clinical trial NCT06609889) (https://clinicaltrials.gov). Unfortunately, there are no reports yet on the beneficial effects of this innovative treatment”.

Why choose patients with Dravet syndrome and GLUT1 deficiency to study specificity of the biomarkers? Shouldn't samples from patients with more common types of epilepsy have been included?

            We wanted to use as many samples from different genetic epilepsies as possible. However, only in the case of Dravet and GLUT1 deficiency samples, we obtained informed consent from the patients to be used in our study.

Please explain why total blood and serum were also not analyzed and, consequently, other types of biomarkers rather than those present in plasma. In fact, in the first sentence of the results section, authors refer “In a recent study, we reported elevated levels of CXCL10, S100B, and CCL20 proteins in the blood serum of a mouse model of LD”

            To perform ELISA assays total blood is not usually recommended. We decided to use plasma instead of serum because it has several advantages: It is faster to obtain, it has a higher yield of possible markers, it has a more complete protein profile, and it offers better reproducibility.

By analyzing Figure S1, it seems to me that a lot of variances occurred between measurements. Can that be explained? Could that be important for further interpretation rather than just analyzing the median?

            The reviewer is right. When we analyzed the samples by ELISA for the mediators indicated in Fig. S1, we obtained a high variability. At present, we do not understand this result. However, in the cytokine profile analysis using the Proteome Profiler Human Cytokine Array kit and a larger sample size, the levels of CXCL10 and CCL20 were not differentially present in LD samples. Therefore, we concluded that CXCL10 and CCL20 were not good biomarkers of LD.

The authors state that PDGF-BB levels in healthy samples showed substantial variability. The same seems to happen in BDNF samples of LD patients. How can that be explained?

            We have no explanation for this fact. In any case, our results indicate that these markers are not as good as EGF and MIF to be used as biomarkers of LD.

In the discussion, care must be taken with abbreviations early mentioned.

            We thank the reviewer very much for pointing out this mistake. We have corrected it accordingly.

Considering that the upmost targets biomarkers that were found altered on LD patients in this study (PDGF-BB, EGF, BDNF, and MIF) are also changed in other neurological diseases, sometimes hardly diagnosed in early stages – Alzheimer’s disease, ALS, Parkinson’s disease, multiple sclerosis - how can these biomarkers be robust for LD?

            Recent reports indicate that neuroinflammation is common among neurological disorders. Our work points out that some growth factors are differentially expressed in the plasma of Lafora disease patients. Perhaps the shortage of these markers is related to the neuroinflammation present in LD, and this could be extrapolated to other neurological disorders. In this manuscript, we are not saying that the biomarkers we have analyzed are specific to Lafora disease. However, the fact that a decrease in the levels of these biomarkers is already detected in asymptomatic patients, allows an early detection of the disease, making possible an early treatment.

Round 2

Reviewer 1 Report

Comments and Suggestions for Authors

I have no additional comments.

Comments on the Quality of English Language

The English could be improved to more clearly express the research.

Reviewer 2 Report

Comments and Suggestions for Authors

The authors properly answer to my concerns and I consider that, now, the manuscript can be published